# Functional Study of *PgGRAS68-01* Gene Involved in the Regulation of Ginsenoside Biosynthesis in *Panax ginseng*

**DOI:** 10.3390/ijms24043347

**Published:** 2023-02-08

**Authors:** Chang Liu, Kangyu Wang, Ziyi Yun, Wenbo Liu, Mingzhu Zhao, Yanfang Wang, Jian Hu, Tao Liu, Nan Wang, Yi Wang, Meiping Zhang

**Affiliations:** 1College of Life Science, Jilin Agricultural University, Changchun 130118, China; 2Jilin Engineering Research Center Ginseng Genetic Resources Development and Utilization, Jilin Agricultural University, Changchun 130118, China; 3Laboratory for Cultivation and Breeding of Medicinal Plants of National Administration of Traditional Chinese Medicine, Jilin Agricultural University, Changchun 130118, China

**Keywords:** *Panax ginseng*, *PgGRAS68-01* gene, ginsenoside biosynthesis, functional genomics, genetic transformation

## Abstract

Ginseng (*Panax ginseng* C. A. Meyer) is a perennial herb from the genus *Panax* in the family Araliaceae. It is famous in China and abroad. The biosynthesis of ginsenosides is controlled by structural genes and regulated by transcription factors. GRAS transcription factors are widely found in plants. They can be used as tools to modify plant metabolic pathways by interacting with promoters or regulatory elements of target genes to regulate the expression of target genes, thereby activating the synergistic interaction of multiple genes in metabolic pathways and effectively improving the accumulation of secondary metabolites. However, there are no reports on the involvement of the *GRAS* gene family in ginsenoside biosynthesis. In this study, the *GRAS* gene family was located on chromosome 24 pairs in ginseng. Tandem replication and fragment replication also played a key role in the expansion of the *GRAS* gene family. The *PgGRAS68-01* gene closely related to ginsenoside biosynthesis was screened out, and the sequence and expression pattern of the gene were analyzed. The results showed that the expression of *PgGRAS68-01* gene was spatio-temporal specific. The full-length sequence of *PgGRAS68-01* gene was cloned, and the overexpression vector pBI121-PgGRAS68-01 was constructed. The ginseng seedlings were transformed by *Agrobacterium rhifaciens*-mediated method. The saponin content in the single root of positive hair root was detected, and the inhibitory role of *PgGRAS68-01* in ginsenoside synthesis is reported.

## 1. Introduction

Medicinal plants are inseparable from people’s lives. They can produce secondary metabolites, which are used for health preservation, treatment, and prevention of diseases. Ginseng is a perennial herb of the genus *Panax* family Araliaceae. All the tissues and parts of *Panax ginseng* can be used as Chinese herbal medicine, and it is a very important medicinal material [1,2]. Ginsenosides are the main bioactive components of ginseng and secondary metabolites of ginseng cells, which have high health benefits and medicinal value. Oleane-type pentacyclic triterpenoid saponins and Dammarane-type tetracyclic triterpenoid saponins are the main types and bioactive components of ginsenosides. Dammarane-type tetracyclic triterpenoid saponins are divided into two groups: 20 s-protopanaxadiol (PPD), such as Ra1, Ra2, Rb, Rb2, and Rb3, and 20S-protopanaxatriol (PPT), such as Re, Rf, Rg1, Rg2, and Rh1 [3]. The biosynthetic pathways of triterpenoid saponins include mevalonate (MVA) and 2-C-methyl-D-erythritol 4-phosphate (MEP). The ginsenoside biosynthesis pathway consists of more than 20 consecutive enzymatic reaction steps. A series of key enzymes, such as 3-hydroxy-3-methylglutaryl CoA reductase (HMGR), farnesyl pyrophosphate synthase (FPS), squalene synthase (SS), and squalene epoxidase [4,5,6,7] play an important role in ginsenoside synthesis, and the study of these key enzymes and their regulatory genes has attracted much attention.

Transcription factors are important regulatory factors affecting the growth and development of higher plants, physiological processes, and network regulation, and are also essential in the regulation of gene expression. *GRAS* transcription factors appeared in land plants through lateral bacterial transfer and were transmitted in the ancestors of bryophytes, lycophytes, and higher plants [8].

GRAS proteins have highly conserved sequences in their carboxyl termini. However, the carboxyl termini of some GRAS proteins contain two conserved GRAS domains or one GRAS domain and one functional region. GRAS proteins contain several ordered motifs LHRI, VHIID, LHRII, PFYRE, and SAW at the carboxyl terminus [9,10]. Among these conserved sequences, the VHIID sequence (where V is valine, I is isoleucine, H histidine, and D is aspartic acid), which is essential for protein–protein interactions, is located midway between the leucine-rich LHR I and LHR II motifs. It is the most essential and central part of GRAS proteins [11,12]. The amino-terminal portion of GRAS proteins is variable except for the highly conserved DELLA protein and TVHYNP protein, both of which are involved in gibberellin signal transduction and can control plant growth and development [13]. This further demonstrates the functional specificity of GRAS proteins.

DELLA protein is a regulator of nuclear transcription, and DELLA protein mainly restricts plant growth by inhibiting GA signal transduction [14,15]. The DELLA domain is a domain unique to DELLA subfamily proteins in the GRAS transcription factor subfamily, and its function is related to GA response. Studies have shown that DELLA protein is a constitutive repressor of GA response and inhibits GA-mediated plant growth and development. GA promotes plant growth and development by reducing the repressor of DELLA proteins. Large-scale genetic screens and analyses in Arabidopsis and rice have identified several key components of GA signaling, including the soluble GA receptor GID1 and the growth repressor DELLA protein [16,17].

Liu et al. found that DELLA proteins in Arabidopsis could interact with JAZ1, a repressor of jasmonate [18]. In plants, JAZ1 inhibits the MYC2 transcription factor and thus affects the production of jasmonate. DELLA protein promotes MYC2 production and enhances its binding to downstream target genes by inhibiting JAZ1, thereby increasing jasmonate production [13]. According to the study by Zhang et al. in 2017, AtDELLA protein can promote anthocyanin synthesis. The anthocyanin synthesis pathway was inhibited in mutants such as gai-t6 and rgl1-1. DELLA protein increases anthocyanin synthesis by inhibiting PAP1 protein [19]. In medicinal plants, the overexpression of *SmSCR1* in *Salvia miltiorrhiza* resulted in significantly higher accumulation of tanshinone than that in the control group [20]. In *Dendrobium officinale*, eight *GRAS* genes were upregulated in different tissues after heat and salt stress [21]. In garlic, *GRAS* gene has five protein-coding genes that have been identified as DELLA samples, three genes *Asa2G00237.1, Asa2G00240.1*, and *Asa4G02090.1* have response to exogenous GA_3_ treatment, and part of the *GRAS* genes regulating garlic bulb growth [22]. In a previous study, the *PgGRAS* gene expression was activated in ginseng hair-like roots treated with gibberellin acid (GAs), and qPCR analysis showed that *PgGRAS* genes of DELLA subfamily played an important role in response to GA treatment in ginseng hairy roots [23]. Therefore, we speculated that *PgGRAS* gene might also play an important role in the biosynthesis of ginsenosides.

In this study, *PgGRAS68-01*, a gene highly related to ginsenoside biosynthesis, was identified based on published *PgGRAS* genes transcription factor data [23]. The PCR cloning method was used to clone *PgGRAS68-01* gene, and its protein sequence characteristics, phylogenetic relationship and expression pattern were analyzed. Then, *Agrobacterium tumefaciens* was used to overexpress *PgGRAS68-01* gene in ginseng explants, and the role of *PgGRAS68-01* gene in ginsenoside synthesis in ginseng was initially revealed. The results of this study can lay a theoretical foundation for further research on the function of *PgGRAS* and the industrial production of ginsenosides through genetic engineering.

## 2. Results

### 2.1. Chromosome Distribution and Gene Replication of PgGRAS Gene

We found that members of the *PgGRAS* gene family showed an uneven distribution on 24 chromosomes of ginseng (Figure 1A). These genes are not found on chromosomes 8, 15, 16, 18, 20, and 24. We noted that *PgGRAS* gene had the highest gene density at different positions on chromosome 1, while there was only one *PgGRAS* gene on chromosome 3, 6, and 13. Interestingly, to investigate *PgGRAS* gene replication events, we found collinear segmented replication of the *PgGRAS* gene in ginseng. This result can prove the existence of *PgGRAS* gene replication in ginseng genome (Figure 1B).

### 2.2. GA Treatment Affects the Synthesis of Ginsenosides

The total saponins of hair roots of ginseng treated with gibberellin were measured. Compared with the untreated hairy root, the total glycosides content of the treated hairy root was higher than that of the untreated hairy root (Figure 2). It can be proven that gibberellin plays a role in inhibiting ginsenoside synthesis in the hair root culture system of ginseng.

### 2.3. Screening of GRAS Candidate Genes Involved in Ginsenoside Biosynthesis

Ginsenoside is the main active component of ginseng, but its content in ginseng is very low, so it is very important to study the synthesis pathway of ginsenoside. A total of 34 genes were found to be significantly correlated with ginsenoside content (Appendix A), among which 18 *PgGRAS* genes were positively correlated with ginsenoside content, and 14 *PgGRAS* genes were negatively correlated with ginsenoside content. Two *PgGRAS* genes (*PgGRAS62-03* and *PgGRAS69-03*) were positively correlated with saponin content, and also negatively correlated with saponin content.

Many key enzymes in ginsenoside synthesis have been cloned and verified. *PgGRAS* genes may be related to key enzyme genes in ginsenoside synthesis pathway. In order to explore the relationship between *PgGRAS* genes and key enzyme genes, SPSS version 23.0 software was used to calculate the correlation between the expression levels of key enzyme genes and *PgGRAS* genes (Appendix A). A total of 34 genes were found to be significantly correlated with the expression of key enzyme genes. Among them, the expression of 19 *PgGRAS* genes was positively correlated with the expression of key enzyme genes. Among them, the expression of 9 *PgGRAS* genes was significantly negatively correlated with the expression of key enzyme genes.

Since the key enzyme genes of ginseng participate in the ginsenoside synthesis pathway and are an important part of it, it is particularly important to study the correlation between the expression of key enzyme genes and ginsenoside content. Therefore, the comparison and analysis of the results calculated by SPSS version 23.0 software found a total of 10 genes (*PgGRAS44-04*, *PgGRAS45-01*, *PgGRAS51-01*, *PgGRAS51-02*, *PgGRAS56-02*, *PgGRAS62-06*, *PgGRAS63-01*, *PgGRAS63-02*, *PgGRAS65-03*, and *PgGRAS68-01*) were significantly correlated with both key enzyme genes and ginsenoside content of ginseng. However, the *PgGRAS68-01* gene was found to be related to more key enzyme genes through network (Figure 3), so *PgGRAS68-01* gene was finally selected for subsequent functional verification.

### 2.4. Sequence Analysis of PgGRAS68-01 Gene

The total length of *PgGRAS68-01* gene is 2041 bp, encoding 540 amino acids with molecular weight of 59,906.01 kDa and isoelectric point (PI) of 5.03. The secondary structure of *PgGRAS68-01* protein contained 257 alpha helix (47.59%), 27 beta turn (5.00%), 202 random coil (37.40%), and 54 extended strand (10.00%) (Figure 4A). Tertiary structural modeling showed that PgGRAS68-01 was composed of alpha helix, beta turn, and random coil (Figure 4B). In order to reveal the evolutionary relationship between *GRAS* genes in different species, the protein sequences of three GRAS family members of Arabidopsis, rice, and tomato were downloaded from NCBI (Appendix A). Phylogenetic trees were constructed using three PgGRAS protein sequences containing PgGRAS68-01 and nine GRAS protein sequences from three other species (Figure 4C). PgGRAS68-01 has the closest evolutionary relationship with Arabidopsis AtGRAS3 and AtGRAS10. At the protein level, PgGRAS68-01 showed a high degree of similarity to its orthologs in different plant species (Figure 4D).

### 2.5. Expression Pattern Analysis of PgGRAS68-01 Gene in Ginseng

In order to understand the expression pattern of *PgGRAS68-01* gene in ginseng, we retrieved the *PgGRAS68-01* gene expression data from 4 different aged stages of ginseng roots, 14 different tissues of 4-year-old ginseng, and 42 farmer’s cultivars of 4-year-old ginseng roots, and plotted the bar graph (Figure 5). In order to more intuitively reflect the expression level of *PgGRAS68-01* gene, we drew the gene expression heat map. Among the 4 different aged stages of ginseng roots, the expression level of *PgGRAS68-01* gene was the highest in 18-year-old ginseng roots and the lowest in 12-year-old ginseng roots (Figure 6A). Among the 14 different tissues of 4-year-old ginseng, *PgGRAS68-01* was expressed in all the tissues, with high expression in fruit flesh and rhizome and the least expression in leaf blade (Figure 6B). Among the 42 farmer’s cultivars of 4-year-old ginseng roots, *PgGRAS68-01* gene expression was significantly different in most of the farmer’s cultivars, with higher expression in S21 and S23 and lower expression in S7 and S36 (Figure 6C).

### 2.6. Cloning and Vector Construction of PgGRAS68-01 Gene

The total RNA of Jilin ginseng was extracted and reverse transcribed into cDNA, which was used as a template to clone *PgGRAS68-01* gene. As shown in Figure 7A, a clear band at 1656bp was observed. The cloned target gene fragment was ligated with T vector and transformed into *Escherichia coli* and screened by blue and white spots. The selected single colonies were amplified and PCR positive clones were verified. The cloned vector was sequenced and the results were identical with the target gene. Plasmids were recovered and double digested with *Xba* I and *Sma* I (Figure 7B). Agarose gel electrophoresis was used for separation and recovery. After the target band was recovered, T4 ligase was used to connect it to the pBI121 expression vector to construct the recombinant plant expression vector plasmid PBI121-PGGras68-01 (Figure 7C), which was transformed and verified by bacterial liquid PCR (Figure 7D). These results indicated that the expression vector pBI121-PgGRAS68-01 was successfully constructed.

### 2.7. Genetic Transformation of Ginseng by Recombinant Plasmid

The recombinant plasmid was transformed into *Agrobacterium* A4 and PCR positive cloning detection was performed; the bacterial solution with clear and bright bands was selected for preservation. The petioles of ginseng sterile seedlings were infected with the transformed Agrobacterium A4 and cultured in 1/2MS medium containing 200 mg·L^−1^ Cef under dark conditions for 5 weeks. Hairy roots were observed on the explants (Figure 8A–D). When the hairy roots grew to about 2 cm, they were transferred to the medium containing 200 mg·L^−1^ Cef for propagation. After obtaining hair-like roots, PCR method was used to detect positive hair-like roots. According to the recombinant plasmid, four primers were designed (Figure 8E), and the genomic DNA of hair-like roots was extracted for PCR detection. The size of *PgGRAS68-01* gene band is shown in Figure 8F and the band is bright and clear. *PgGRAS68-01* gene integration in the hair root genome can be preliminarily identified.

### 2.8. Detection of Ginsenoside Content in Positive Ginseng Hairy Roots

The single root system of ginseng hair-like root with positive test was propagated in large numbers (Figure 9A), and the fresh ginseng hair-like root (Figure 9B) was dried to obtain the dry ginseng hair-like root (Figure 9C). The methanol extract of positive hair-like root was obtained by grinding the dry ginseng hair-like root and extracting with Soxhlet extractor. The content of ginsenoside in positive hair-like roots was detected by HPLC, and the results showed that the sum content of ginsenoside (Re + Rf + Rb1 + Rc + Rb2 + Rd + Rg3 + Rh2 + PPT + PPD) in the single root of positive hair-like roots of transgenic line-18, transgenic line-19, and transgenic line-30 was lower than that of the control group (Figure 10A), indicating that the transfer of *PgGRAS68-01* gene had an inhibitory effect on ginsenoside. This was also consistent with the previous results that *PgGRAS68-01* gene was significantly negatively correlated with the content of saponins. As shown in Figure 10B–G, analysis of monomeric saponins showed that compared with the control group, the contents of monomeric saponins Rg1, Rb1 and Rg3 were significantly reduced, PPD and PPT were also significantly reduced, but the content of monomeric saponins Re was increased. These results demonstrate that *PgGRAS68-01* regulates ginsenoside synthesis.

## 3. Discussion

To date, GRAS transcription factors have been studied in many species and have been identified in plants such as *Arabidopsis thaliana* (33) [12] (Lee et al., 2008), rose (59) [24], rice (57) [25], tomato (53) [26], and grape (52) [27]. The *GRAS* gene family of *Arabidopsis thaliana*, a model plant, has been well studied, and the functions of several family members have been verified. We found that most of the ginseng *GRAS* gene family was distributed on 24 pairs of chromosomes, and only a few chromosomes did not have *PgGRAS* gene. *PgGRAS* gene duplication occurred in 24 pairs of ginseng chromosomes, and a total of 11 duplicate events were identified (Figure 1). Meanwhile, *GRAS* gene duplications have been found in many species, including quinoa [28], switchgrass [29], millet [30], sorghum [31] and alfalfa [32]. In Arabidopsis, DELLA proteins can inhibit plant growth by inhibiting the GA pathway [17]. Ginsenosides are the main active components of ginseng and have high medicinal value. Previous studies showed that the dry weight and fresh weight of hair roots of ginseng decreased after exogenous addition of GA, indicating that the addition of GA inhibited its growth [23]. In order to explore the effect of exogenous GA on ginsenosides content, different concentrations of GA were used to treat ginseng hair-like roots. The results showed that the total glycosides content of all treated ginseng hair-like roots was lower than that of untreated ginseng hair-like roots (Figure 2). The results showed that exogenous GA could inhibit the growth and development of ginseng and the synthesis of ginsenoside.

*GRAS* genes are involved in the regulation of anthocyanins and other secondary metabolites [19], but studies on *GRAS* genes involved in the biosynthesis of ginsenosides have not been reported. In the *PgGRAS* gene family, *PgGRAS68-01*, which is highly correlated with ginsenoside content, belongs to the DELLA subfamily. Sequence alignment and evolutionary analysis with other species revealed that *PgGRAS68-01* had the closest evolutionary relationship with Arabidopsis *AtGRAS10* and rice *OsGRAS25*, and both had DELLA domains at the N-terminus, which was consistent with our previous study. Interestingly, after analyzing the expression pattern of *PgGRAS68-01* gene, we found that *PgGRAS68-01* was expressed in all the 42 farmer’s cultivars, indicating that the expression of *PgGRAS68-01* was widespread. However, *PgGRAS68-01* also had its specificity and was highly expressed in S21 and S23 farm cultivars. The expression of *PgGRAS68-01* gene in 14 different tissues of 4-year-old ginseng was also specific, and the highest expression level was found in fruit flesh. The expression level of *PgGRAS68-01* was the highest in the roots of 4 different aged stages of ginseng roots, which indicated that the expression of *PgGRAS68-01* gene in ginseng was spatio-temporal specific.

Then, we cloned the full-length *PgGRAS68-01* gene and constructed a recombinant plant overexpression vector. After genetic transformation of ginseng explants, positive hairy roots were finally obtained. Detection of ginsenoside content in single root of positive hairlike root showed that *PgGRAS68-01* gene inhibited ginsenoside production, which also proved that *PgGRAS68-01* gene regulated ginsenoside synthesis. Subsequently, some monomeric saponins were analyzed, and it was found that the content of monomeric saponins such as Rg1, Rb1, and Rg3 decreased, while the content of monomeric saponins Re increased (Figure 10). Through the study of ginsenoside synthesis pathway, it was found that monomeric saponin Re was generated from monomeric saponin Rg1 in the original ginsentriol saponin synthesis pathway [3]. However, the content of monomeric saponin Re increased while the content of monomeric saponin Rg1 decreased in the single root of positive hair roots. Therefore, it is speculated that *PgGRAS68-01* can promote the conversion of a large amount of Rg1 to Re, so the content of Rg1 decreases while the content of Re increases.

In conclusion, in this paper we believe that under the action of gibberellin, the expression of *PgGRAS68-01* gene increases, thus inhibiting ginsenoside synthesis (Figure 11). Previous studies have found that the generation of monomeric saponins is regulated by key enzyme genes in the ginsenoside synthesis pathway [4,5], and previous studies have shown that transcription factors generally affect metabolic pathways through the regulation of structural genes. Therefore, it is speculated that *PgGRAS68-01* gene affects the content of ginsenoside by acting on key enzyme genes in the ginsenoside synthesis pathway, but this part remains to be studied.

## 4. Materials and Methods

### 4.1. Data and Materials

The data sources used in this study were all from the Jilin Ginseng Unigenes Database constructed by our laboratory [33]. The database was obtained by RNA extraction and sequencing of 14 tissues of four-year-old Jilin ginseng cultivar Damaya. Then, with 248,993 Unigenes constructed as a reference, Trinity software (version 2013-02-25) was used to determine the expression levels of 25-, 18-, 12-, and 5-year-old roots of ginseng, 14 tissues of ginseng, and 42 farmer’s cultivars of ginseng. Expression was calculated using Transcripts Per Million (TPM). In addition, the genome derived from ginseng [34](Wang et al., 2022) was also used in this study. The ginseng receptor material used in this experiment is sterile ginseng seedlings generated from ginseng seeds kept in the laboratory. All strains and vectors used were kept in the laboratory.

### 4.2. PgGRAS Gene Duplication and Chromosome Localization

Our laboratory previously identified *PgGRAS* genes in Jilin Ginseng Unigenes Database [23]. We used Blastn to compare the *PgGRAS* genes identified with ginseng genome [34] to determine their distribution in the ginseng genome and the variation of *PgGRAS* gene family size among different genotypes. With identity ≥ 99% and cover length ≥ 300 bp as the comparison standard, the distribution of genes of this gene family in ginseng genome was compared with each genome combination. The R-Package Circlize structure of this gene was used to determine the pan-and core transcripts of this gene family in the ginseng genotypes. The position of the transcript on the chromosome was visualized using the MG2C online tool (http://www.mg2c.iask.in/mg2c_v2.1/index.html, accessed on 1 November 2021).

### 4.3. Culture of Ginseng Hair Roots Induced by Gibberellin and Analysis of Saponin Content Change

Ginseng hairy roots were treated with gibberellin according to the method in Wang et al. at 2020 year. The hair roots of ginseng treated with gibberellin were dried in an oven at 37 °C, and then the dried hair roots were ground to powder, wrapped in filter paper, and soaked in methanol overnight. Ultrasound-assisted method was used for 30 min at 60 °C. The saponins were extracted in a Soxhlet extractor, and the methanol was evaporated in a water bath at 60 °C using a rotary evaporator and redissolved in 20 mL distilled water. The aqueous solution was placed in a separate funnel and extracted three times with an equal volume of ethyl acetate, and the ester phase was removed. The water layer was extracted three times with an equal volume of saturated n-butanol. The n-butanol phase was recovered to a round-bottom flask and evaporated again at 90 °C using a rotary evaporator, Dissolve in 10 mL methanol and set aside. The 40 μL sample solution was dried in a water bath, and the freshly prepared 0.2 μL vanillin (50 mg/mL)–glacial acetic acid solution and 800 μL perchloric acid were added. The solution was heated in a constant temperature water bath at 60 °C for 15 min, and then cooled by running water for 5 min. Then, 5 mL glacial acetic acid was added and the solution was shaken well immediately. The accompanying reagent was used as a blank control, and the absorbance was measured at 544 nm with a microplate reader.

### 4.4. Identification of PgGRAS Genes Related to Ginsenoside Synthesis

The expression data of *PgGRAS* genes in 42 farmer’s cultivars of Jilin ginseng and the saponin content data in 42 farmer’s cultivars of Jilin ginseng were sorted out, and then Pearson correlation coefficient was analyzed by SPSS software version 23.0. SPSS Statistics version 23.0 software was used to calculate the correlation between ginsenoside content and *PgGRAS* genes expression in 42 farmer’s cultivars (Appendix A). The expression data of *PgGRAS* genes in 42 farmer’s cultivars of Jilin ginseng were extracted and placed in the same table with the known expression levels of key enzyme genes for ginsenoside synthesis. Pearson correlation coefficient was also analyzed by SPSS version 23.0 software. Spearman’s correlation coefficients were calculated using R programming language version 3.5.0 and software (http://www.rproje ct.org/, accessed on 1 December 2021). BioLayout Express ^3D^ version 3.2 software was used to construct the co-expression network of *PgGRAS* gene and key enzyme genes of ginsenoside biosynthesis and complete the visualization of gene interaction network.

### 4.5. Characteristic Analysis of PgGRAS68-01 Gene

Online software Expasy ProtParam (https://web.expasy.org/protparam/, accessed on 10 December 2021) was used to predict PgGRAS68-01 the basic physical and chemical properties of protein, including the theory of relative molecular mass (KDa) and isoelectric point (PI). Using the SOPMA (https://npsa-prabi.ibcp.fr/cgi-bin/npsa_automat.pl?page=npsa_sopma.html, accessed on 14 December 2021) [35] and the SWISS-MODEL (https://www.swissmodel.expasy.org/, accessed on 15 December 2021) [36], the secondary and tertiary structure of the protein based on the amino acid sequence of PgGRAS68-01 were analyzed. Phylogenetic trees were constructed with the neighbor joining method of MEGA-X [37] using the protein sequences of PgGRAS68-01 and those of other species, and the guidance repeats were set to 1000. The protein sequences of PgGRAS68-01 and the other three species were aligned using DNAMAN version 5.2.2 software.

### 4.6. Expression Analysis of PgGRAS68-01 Gene

In order to further analyze the expression pattern of *PgGRAS68-01* in ginseng, we selected the expression level of *PgGRAS68-01* in 4 different aged stages (5, 12, 18, 25 years old) of ginseng roots, 14 different tissues of 4-year-old ginseng, and 42 farmer’s cultivars of 4-year-old ginseng roots from the previous research data of the laboratory [23]. The expression was visualized using the TBtools version 1.096 software [38].

### 4.7. Cloning of PgGRAS68-01 Gene

Total RNA of ginseng was extracted by TRIZOL method and reverse transcribed into cDNA. The most appropriate primers were designed according to the basic principles of primer design, and the appropriate restriction sites were screened out in the target genes according to the vector map of pBI121. *Xba*I and protective bases were added at the upstream 5′ end, and *Sma*I restriction sites were added at the downstream 5′ end of the primers. Primers as follows, F: 5′-TGCTCTAGAATGTGGGAGGAAACTGAACAAG, R: 5′-TCCCCCGGGGAGGTCCACCACCAACTGAGT. The cDNA obtained by reverse transcription was used as the template for PCR amplification. The reaction system was as follows: 5 μL 2 × Master Mix, 0.5 μL Sense primer, 0.5 μL Anti primer, 1 μL cDNA, 3 μL ddH_2_O, for a total of 10 μL. The target gene fragment was obtained for use.

### 4.8. Construction of Vector and Genetic Transformation of Ginseng

Using TIANGEN PGM-T cloning ligation kit, the target gene fragment was ligated with the cloning vector and transferred into the competent *Escherichia coli* DH5α cells for culture. The sequencing results of the positive single colony were compared with the nucleotide sequence of the target gene. If the two were identical, the cloning vector *PgGRAS68-01* was successfully constructed. Remember the pGM-T-PgGRAS68-01. The *PgGRAS68-01* plasmid and pBI121 plasmid were double digested with *Xba*Ⅰ and *Sma*Ⅰ restriction enzymes, and the target gene of *PgGRAS68-01* was ligated with the expression vector pBI121 by TAKALA T4 DNA ligase, and the double digestion was verified. The candidate gene overexpression vector pBI121-PgGRAS68-01 was successfully constructed.

To prepare genetically engineered bacteria, the pBI121-PgGRAS68-01 overexpression vector was constructed to transform competent *Agrobacterium* A4 cells. The petioles and roots of ginseng seedlings were precultured on MS solid medium containing 2, 4-D + 6-BA. After pre-feeding, cut into small pieces and place them in the resuspended A4 bacterial solution for infection. The infected explants were placed on solid MS medium containing acetosyringone (AS). After 3 days of co-culture at 22 °C in the dark, the explants were transferred to 1/2MS solid medium containing Cef and cultured at 22 °C in the dark, and the growth was observed and recorded. Primers were designed for the expression vectors successfully connected to the target gene, and these primers were used for PCR positive verification. The single root system of hair root which was positive was propagated.

### 4.9. Ginsenoside Content of Positive Hair Roots

According to the above method of extracting saponins, extract the positive hair root ginsenoside for use. The content of ginsenoside in positive hair roots was detected by HPLC with Waters e2695 HPLC on a Waters C18 column (4.6 × 250 mm, 5 μM) at 35 °C with an injection volume of 20.0 μL. Mobile phase: A. water, B. acetonitrile, gradient elution (Table 1), the flow rate of mobile phase was 1.0 mL/min, and the detection wavelength was 203 nm. Preparation of standard solution: ginsenoside standards Rb1, Rb2, Rb3, Rc, Rd, Re, Rf, Rg1, Rg2, Rg3, PPD, and PPT were accurately weighed, and an appropriate amount of methanol was added for later use.

## 5. Conclusions

In this study, most *PgGRAS* genes were located on 24 pairs of chromosomes in ginseng, and gene duplication was observed. The correlation analysis between the expression of *PgGRAS* genes and the content of saponins and the expression of key enzyme genes showed that 34 *PgGRAS* were significantly correlated with the content of monomeric saponins and 34 *PgGRAS* were significantly correlated with the expression of key enzyme genes. There were 10 genes that were significantly correlated with both saponin content and key enzyme gene expression. The *PgGRAS68-01* gene was found to be associated with more key enzyme genes by network, indicating that *PgGRAS* genes were directly or indirectly involved in the regulation of ginsenoside synthesis. By analyzing the sequence of *PgGRAS68-01* gene, we found that it has similar structure to *GRAS* gene of other species. After analyzing the expression pattern of *PgGRAS68-01*, we found that *PgGRAS68-01* had spatiotemporal specificity. Then, we cloned *PgGRAS68-01* gene, constructed plant expression vector, and transformed ginseng explants by Agrobacterium transformation method. After preliminary PCR identification, positive hairy roots were obtained. Compared with the control group, the contents of monomeric saponins Rg1, Rb1, Rg3, PPD, and PPT were significantly reduced, but the content of monomeric saponin Re was increased, indicating that *PgGRAS68-01* gene regulates ginsenoside synthesis.

## Figures and Tables

**Figure 1 ijms-24-03347-f001:**
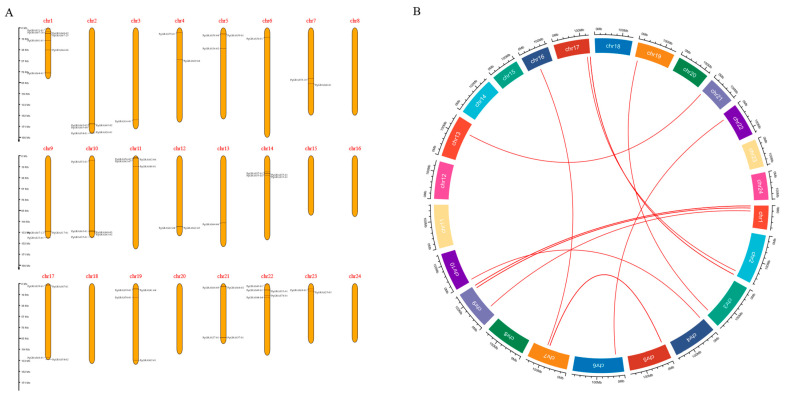
Chromosomal localization and synteny block of the *PgGRAS* gene family in *Panax ginseng*. (**A**) Chromosomal localization of the *GRAS* gene family in *Panax ginseng*. (**B**) Synteny block of *PgGRAS* gene family members within the ginseng genome. Red arcs indicate synteny between genes, Chr: Chromosome, extrachromosomal scale represents the length of chromosome (Mb).

**Figure 2 ijms-24-03347-f002:**
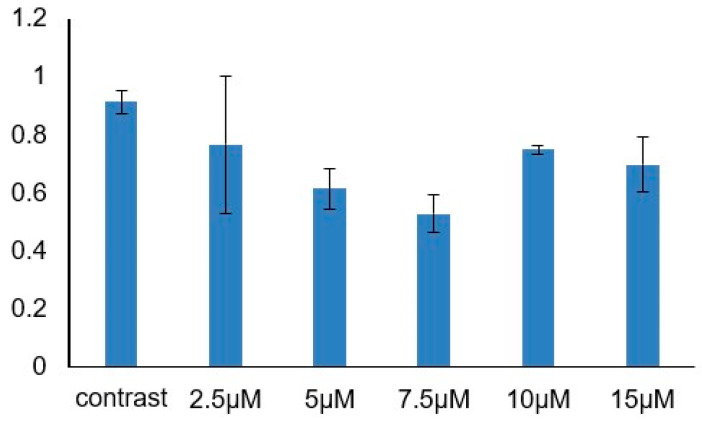
Ginsenoside content of ginseng hairy roots were treated with different concentrations. The abscissa is the concentration of GA and the ordinate is the content of total ginsenosides.

**Figure 3 ijms-24-03347-f003:**
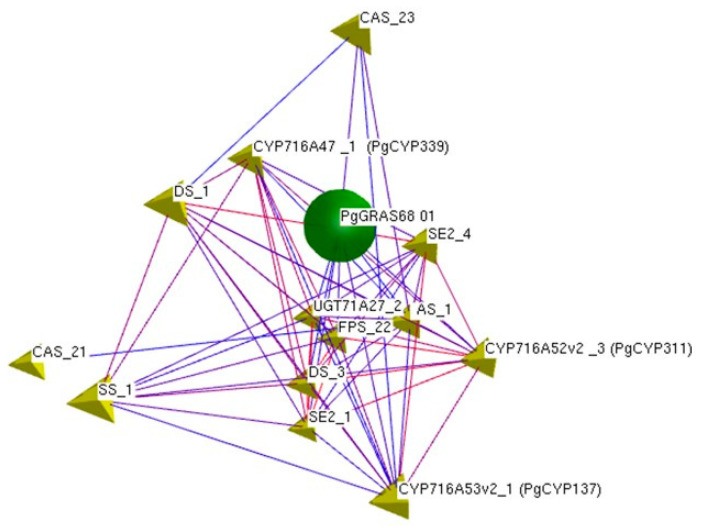
Network interaction between *PgGRAS68-01* gene and key enzyme genes of ginsenoside biosynthesis. The green ball is *PgGRAS68-01* gene, the yellow triangle is key enzyme gene.

**Figure 4 ijms-24-03347-f004:**
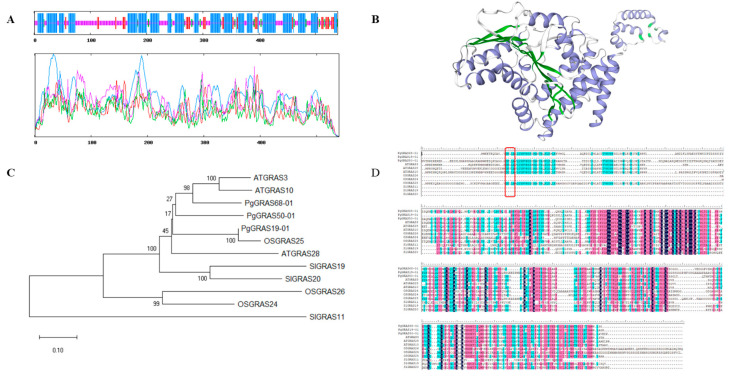
Characterization analysis of the *PgGRAS68-01* gene. (**A**) Secondary structure analysis of PgGRAS68-01 protein in ginseng. The blue, green, purple, and red lines represent alpha helix, beta turn, random coil, and extended strand. (**B**) Tertiary structure of PgGRAS68-01 protein. Purple represents the Alpha helix, green represents the Beta turn, gray represents the Random coil. (**C**) Evolutionary relationships between *PgGRAS68-01* and other members of the *GRAS* gene family and *GRAS* genes in other species. (**D**) Amino acid sequence alignment of PgGRAS68-01 to protein sequences of other species. The red box is the DELLA domain.

**Figure 5 ijms-24-03347-f005:**
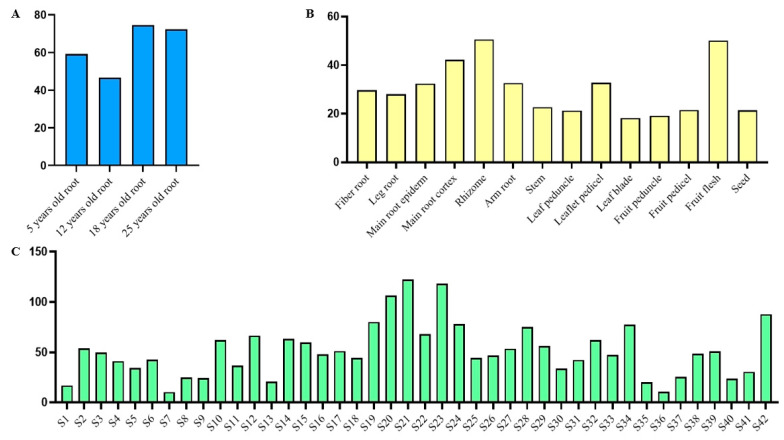
Expression of *PgGRAS68-01* gene in ginseng. (**A**) Expression of *PgGRAS68-01* gene in the 4 different aged stages of ginseng roots. (**B**) Expression of *PgGRAS68-01* gene in 14 different tissues of 4-year-old ginseng. (**C**) Expression of *PgGRAS68-01* gene in the 42 farmer’s cultivars of 4-year-old ginseng roots.

**Figure 6 ijms-24-03347-f006:**
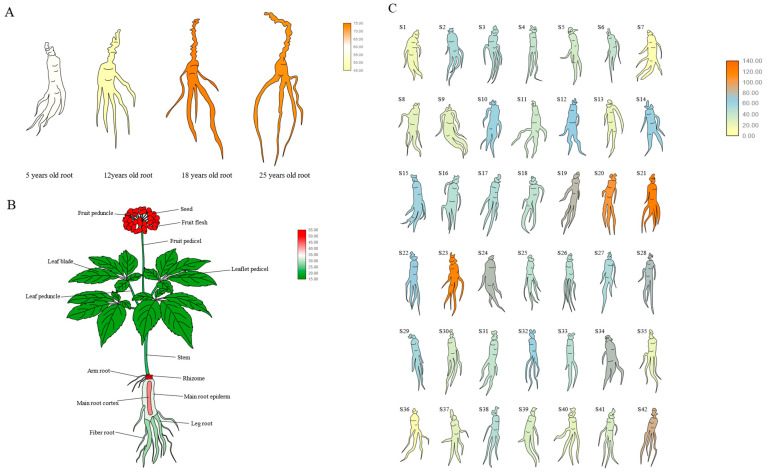
Heatmaps analysis spatiotemporal expression patterns of *PgGRAS68-01* gene in *Panax ginseng*. (**A**) The *PgGRAS68-01* gene expressed in the 4 different aged stages (5, 12, 18, 25 years-old) of ginseng roots. (**B**) The *PgGRAS68-01* gene expressed in the 14 different tissues of 4-year-old ginseng. (**C**) The *PgGRAS68-01* gene expressed in the 42 farmer’s cultivars of 4-year-old ginseng roots.

**Figure 7 ijms-24-03347-f007:**
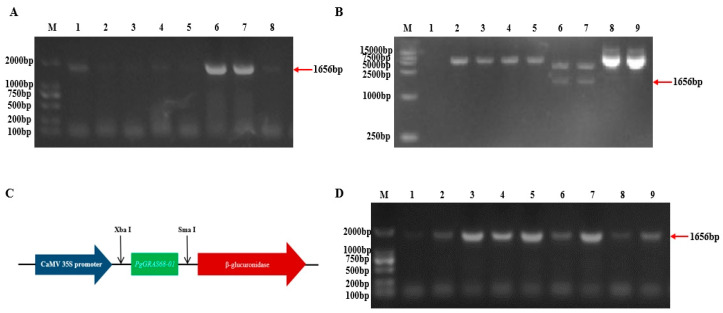
Cloning and vector construction of *PgGRAS68-01* gene. (**A**) The agarose electrophoresis of PCR in ginseng of *PgGRAS68-01*. M: Maker; 1–8: the result of PCR. (**B**) The electrophoretogram of *PgGRAS68-01* gene digestion and cloning vectors. M: Marker; 1: blank control; 2–5: Single digestion; 6,7: Double digestion; 8,9: plasmid. (**C**) Construction of expression vector pBI121-PgGRAS68-01. (**D**) The PCR electrophoretogram of expression vector of *PgGRAS68-01* gene. M: Maker; 2–10 the result of PCR.

**Figure 8 ijms-24-03347-f008:**
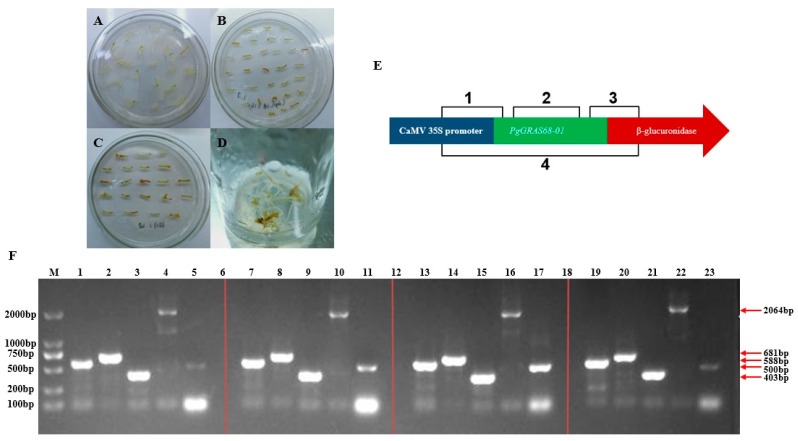
Parameter explants were transformed with recombinant plasmids. (**A**) Leaf handle and root preculture. (**B**) Coculture with *Agrobacterium tumefaciens*. (**C**) Positive hairy roots. (**D**) Culture of positive hairy roots. (**E**) Design of primers for PCR positive validation. (**F**) PCR method tests for some positive ginseng hairy roots strains. M: Maker; 1–5, 7–11, 13–17, 19–23: Positive ginseng hairy roots.

**Figure 9 ijms-24-03347-f009:**
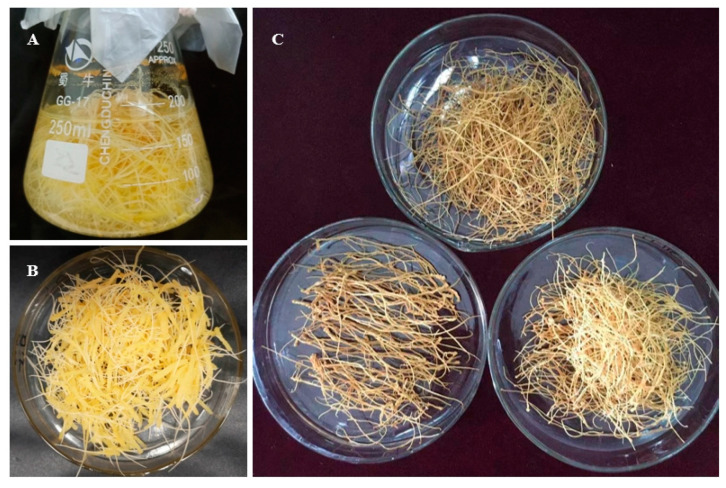
Extraction of ginsenoside from positive ginseng hairy roots. (**A**) Cultivation of single roots of positive ginseng hairy roots using liquid medium. (**B**) Fresh samples of partially positive ginseng hairy roots. (**C**) Partially positive hairy stem samples.

**Figure 10 ijms-24-03347-f010:**
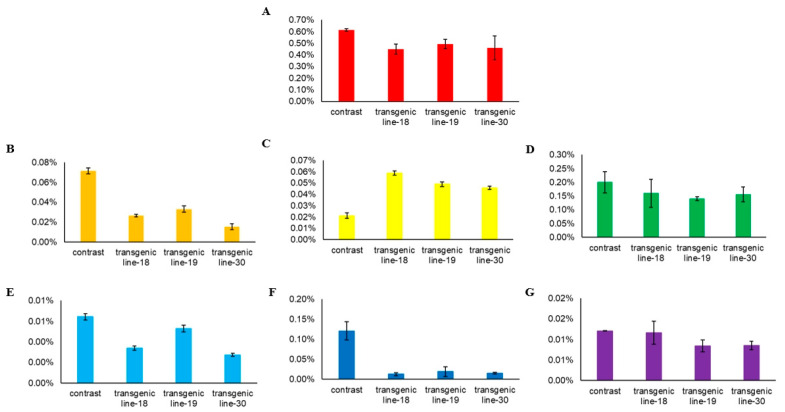
Detection of ginsenoside content in positive hairy roots. (**A**) Ginseng saponin content detection of positive hairy roots single root system transgenic line-18, transgenic line-19, and transgenic line-30. (**B**–**G**) Contents of monomeric saponins Rg1, Re, Rb1, Rg3, PPD, and PPT in transgenic line 18, 19, and 30 of positive hairy roots single root system.

**Figure 11 ijms-24-03347-f011:**
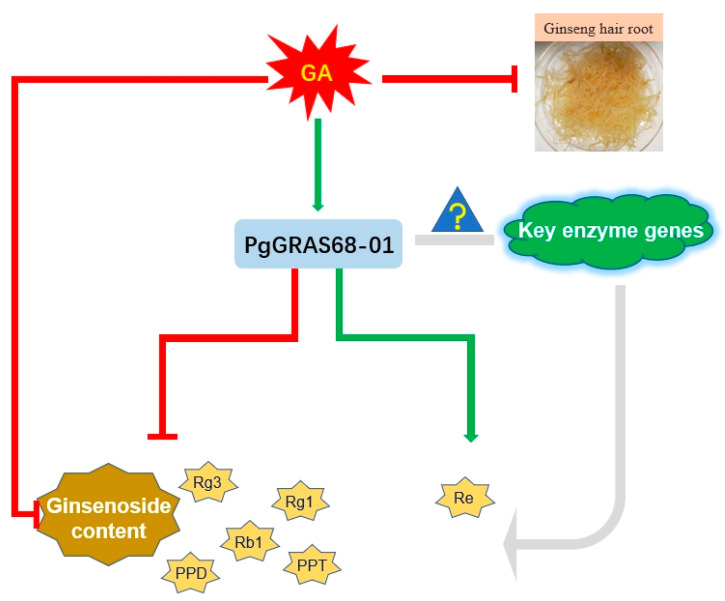
Mechanism of *PgGRAS68-01* gene involved in ginsenoside synthesis.

**Table 1 ijms-24-03347-t001:** Mobile phase gradient condition table for HPLC.

Time (min)	Solvent A (%)	Solvent B (%)	Velocity (mL/min)
0–40	18–21	82–79	1
40–42	21–26	79–74	1
42–46	26–32	74–68	1
46–66	32–33.5	68–66.5	1
66–71	33.5–38	66.5–62	1
71–86	38–65	62–35	1
86–91	65	35	1
91–96	65–85	35–15	1
96–103	85	15	1
103–105	85–18	15–82	1
105–106	18	82	1

## Data Availability

These ginseng sample materials can be accessed upon request to the corresponding author.

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
