# Peer review of "Functional Study of PgGRAS68-01 Gene Involved in the Regulation of Ginsenoside Biosynthesis in Panax ginseng"

_ijms, 2023, doi:10.3390/ijms24043347_

Round 1
Reviewer 1 Report
Some conclusions are overly strong, for instance the observation of duplicated segments does not necessarily prove WGD.
the biggest result here is the correlation between PgGRAS genes and saponins, unfortunately this table is entirely illegible. According to the authors there are highly correlated PgGRAS, but I can't tell.
It's not clear why PgGRAS-01 was chosen for further investigation, but it's expression was examined in existing data where expression levels very by as little as 2-fold, and are very low in some cultivars. Not really clearly causal for ginsenoside production.
In the transgenic lines, where protein production level is unknown, tgere are some differences in saponin content, but only one is dramatically up, Re, and on dramatically down, PPD. It is not clear whether the differences, which with the two exceptions above are small, is due to the introduced gene or a secondary effect. Overall, it's not clear to me that PgGRAS68-01 is a critical regulatory gene, or a minor modulator.
Tab le S1 is very hard to read due to carriage returns between the minus signs and values - needs to be reformatted. Not clear three significant figures are needed,, fewer would be more legible. Perhaps this table could be presented as a heat map.
Author Response
Some conclusions are overly strong, for instance the observation of duplicated segments does not necessarily prove WGD.
Response: Firstly, I want to sincerely thank you for taking the time and attention to review our manuscript. We made a mistake in our statement. It is an intuitive method to identify WGD by collinear analysis of genome. In the whole genome of ginseng, we found that PgGRAS gene could be matched to several different intervals, which proved the existence of PgGRAS gene replication in ginseng genome, but could not prove the existence of WGD. Thank you for your suggestion, we have made the changes.
The biggest result here is the correlation between PgGRAS genes and saponins, unfortunately this table is entirely illegible. According to the authors there are highly correlated PgGRAS, but I can't tell.
Response: Thank you for your suggestion. In the era of big data, correlation coefficient is widely used in life science research. Jahagirdar S, Saccenti E. (doi: 10.3390/metabo10040171) and Dong et al. (doi: 10.3389/fphar.2021.705498) also used the correlation analysis method in their paper research. The two variables for correlation analysis in this study were PgGRAS gene expression levels and ginsenoside expression levels in 42 farmer’s cultivars. SPSS software was used to analyze the two expression levels, so as to measure the degree of close correlation between the two variables. The figures in Table 1S represent P values, which are correlated when P < 0.05. We obtained PgGRAS gene which was significantly related to ginsenoside expression. Similarly, we analyzed the PgGRAS gene expression levels and the key enzyme gene of ginsenoside biosynthesis expression levels in 42 farmer’s cultivars by Pearson correlation coefficient, and obtained PgGRAS gene which was significantly related to the key enzyme gene of ginsenoside biosynthesis.
It's not clear why PgGRAS-01 was chosen for further investigation, but it's expression was examined in existing data where expression levels very by as little as 2-fold, and are very low in some cultivars. Not really clearly causal for ginsenoside production.
Response: Thank you for your suggestion. We did some bioinformatics analysis to get the candidate genes. Jiang et al. (doi: 10.1186/s12859-022-04732-9) and Langfelder P, Horvath S (doi: 10.1186/1471-2105-9-559) showed that, Correlation patterns of gene expression, also known as correlation networks, are usually represented as indirect graphs, where genes are nodes and inter-gene correlations are edges. Edges in correlation networks are defined by correlation measures, such as Pearson correlation coefficient (PCC) and Spearman correlation coefficient (SCC). In order to obtain important genes from related networks, a common method is to extract network modules. A network module on a gene-related network is a group of genes that have high internal correlation but are relatively isolated from the rest of the network. A set of genes that form a network of modules suggests that they may be regulated by similar mechanisms and contribute to common biological functions. Therefore, we used R language to analyze the correlation between PgGRAS gene obtained in 2. And key enzyme genes of ginsenoside synthesis through Pearson correlation coefficient, and drew the network. Through increasing P value, the final result showed that PgGRAS68-01 gene was also closely related to other key enzyme genes. Therefore, we identified PgGRAS68-01 gene for further research.
As for the expression level of PgGRAS68-01 gene, we intended to prove that it was expressed in all farmer’s cultivars through the expression amount, but the expression level was uneven, indicating that the PgGRAS68-01 gene expression is regional and may be affected by geographical environment.
In the transgenic lines, where protein production level is unknown, tgere are some differences in saponin content, but only one is dramatically up, Re, and on dramatically down, PPD. It is not clear whether the differences, which with the two exceptions above are small, is due to the introduced gene or a secondary effect. Overall, it's not clear to me that PgGRAS68-01 is a critical regulatory gene, or a minor modulator.
Response: Thank you for your suggestion. Your concerns are also our concerns. However, we have used our data analysis to obtain our identified candidate gene, and we have been able to verify the accuracy of our analysis by studying the function of the gene through transgenesis. We are still working on the protein level, but at the moment I can identify the PgGRAS68-01 gene as a negatively regulated transcription factor and we are also doing gene editing or RNA interference to verify the function of the gene. All our data analysis suggests that PgGRAS68-01 is an important candidate regulatory gene.
Tab le S1 is very hard to read due to carriage returns between the minus signs and values - needs to be reformatted. Not clear three significant figures are needed, fewer would be more legible. Perhaps this table could be presented as a heat map.
Response: Thank you for your suggestion, we have made the changes.
Reviewer 2 Report
I read and reviewed the manuscript entitled "Functional study of PgGRAS68-01 gene involved in the regulation of ginsenoside biosynthesis in Panax ginseng". I suggest some minor modifications, please, see below.
Best,
Line#
7:… Research Center Ginseng Genetic…Research Center for Ginseng Genetic.. Please, check.
6-8, 10:City, Province Zipcode, Country for consistency. This order is different, see Affiliation 1 & 3.
14: …the genus Panax…the genus Panax..(Italicize the genus name). Also, do the same in line #36.
28-29: …and it was found that PgGRAS68-01 played an inhibitory role in ginsenoside synthesis. … and the inhibitory role of PgGRAS68-01 in ginsenoside synthesis was reported.
37: .. medicine, and it is a very rare…What does it refer to? Panax? Ginseng in general? Specify.
39: which have high health care and medicinal value…which have high medicinal value (Delete: “health care”), it does not help the meaning. You can also replace it with “..health benefits and medicinal value” or so.
49: .. (Kim et al. 2009; Yao et al. 2020; Yonekura-Sakakibara and Hanada 2011; Zheng et al. 2019).. Sort the cited ref here alphabetically or chronological order.
80: … LIU et al. found that … Liu et al. (Year) found that.. Uppercase onle the initial letter of author’s name, also add the year of publication.
84-85: ..According to the study of Zhang et al., AtDELLA protein can… Add the year after “et al.”, You cited Zhang in several years, i.e., 2017, 2021, and 2022.
88: salvia miltiorrhiza … Salvia miltiorrhiza, uppercase “S”, also italicize the scientific name.
89-90: Italicize the scientific name (Dendrobium officinale).
91: has five protein-coding genes have been …has five protein-coding genes that have been…
93: GA3… Type “3” below the line of text (Subscript).
94: study, we found that PgGRAS … study, the PgGRAS…
97: ginseng hairs roots … ginseng hairy roots..
103-104: Agrobacterium tumefaciens… Italicize the scientific name.
136-138: SPSS Sta- 136 tistics software was used to calculate the correlation between ginsenoside content and 137 PgGRASs gene expression in 42 farmer’s cultivars (Table S1). … Move to Materials and Methods section.
141-142: Two PgGRAS genes were positively cor- 141 related with saponin content and negatively correlated with saponin content… Please, check.
191: Adjust the table to fit its content for better demonstration, i.e., column width, ..etc.
206: Add SD or SEM bars to the charts.
259: Check spaces.
281: Check the cited reference style of the journal, also it has to be consistent in the entire manuscript, “(59) (Kumari et al.), Arabidopsis Thaliana (33) (Lee et al. 281
2008), rice (57) (Tian et al. 2004), tomato (53) (Huang et al. 2015)”, these are mixed styles.
Please, check and correct in the entire manuscript.
367: water, The aqueous solution… water. The aqueous solution…
481: Reference list: Check res. Titles (Sometimes all initial uppercases), also check journal names. Italicize scientific names in the entire manuscript including the ref list.
Author Response
I read and reviewed the manuscript entitled "Functional study of PgGRAS68-01 gene involved in the regulation of ginsenoside biosynthesis in Panax ginseng". I suggest some minor modifications, please, see below.
Best,
Response: Firstly, I want to sincerely thank you for taking the time and attention to review our manuscript. Thank you for your suggestion, we have made the changes.
Line#
7:… Research Center Ginseng Genetic…Research Center for Ginseng Genetic.. Please, check.
Response: Thank you for your suggestion, we make sure Research Center Ginseng Genetic is fixed.
6-8, 10:City, Province Zipcode, Country for consistency. This order is different, see Affiliation 1 & 3.
Response: Thank you for your suggestion, we have made the changes.
14: …the genus Panax…the genus Panax..(Italicize the genus name). Also, do the same in line #36.
Response: Thank you for your suggestion, we have made the changes.
28-29: …and it was found that PgGRAS68-01 played an inhibitory role in ginsenoside synthesis. … and the inhibitory role of PgGRAS68-01 in ginsenoside synthesis was reported.
Response: Thank you for your suggestion, we have made the changes.
37: .. medicine, and it is a very rare…What does it refer to? Panax? Ginseng in general? Specify.
Response: Thank you for your suggestion, we have changed to …All the tissues and parts of Panax ginseng can be used as Chinese herbal medicine.
39: which have high health care and medicinal value…which have high medicinal value (Delete: “health care”), it does not help the meaning. You can also replace it with “..health benefits and medicinal value” or so.
Response: Thank you for your suggestion, we have made the changes it.
49: .. (Kim et al. 2009; Yao et al. 2020; Yonekura-Sakakibara and Hanada 2011; Zheng et al. 2019).. Sort the cited ref here alphabetically or chronological order.
Response: Thank you for your suggestion, we have changed the reference.
80: … LIU et al. found that … Liu et al. (Year) found that.. Uppercase onle the initial letter of author’s name, also add the year of publication.
Response: Thank you for your suggestion, we have changed the reference.
84-85: ..According to the study of Zhang et al., AtDELLA protein can… Add the year after “et al.”, You cited Zhang in several years, i.e., 2017, 2021, and 2022.
Response: Thank you for your suggestion, we have changed to Zhang et al. at 2017 year.
88: salvia miltiorrhiza … Salvia miltiorrhiza, uppercase “S”, also italicize the scientific name.
Response: Thank you for your suggestion, we have made the changes.
89-90: Italicize the scientific name (Dendrobium officinale).
Response: Thank you for your suggestion, we have made the changes.
91: has five protein-coding genes have been …has five protein-coding genes that have been…
Response: Thank you for your suggestion, we have made the changes.
93: GA3… Type “3” below the line of text (Subscript).
Response: Thank you for your suggestion, we have made the changes.
94: study, we found that PgGRAS … study, the PgGRAS…
Response: Thank you for your suggestion, we have made the changes.
97: ginseng hairs roots … ginseng hairy roots..
Response: Thank you for your suggestion, we have made the changes.
103-104: Agrobacterium tumefaciens… Italicize the scientific name.
Response: Thank you for your suggestion, we have made the changes.
136-138: SPSS Sta- 136 tistics software was used to calculate the correlation between ginsenoside content and 137 PgGRASs gene expression in 42 farmer’s cultivars (Table S1). … Move to Materials and Methods section.
Response: Thank you for your suggestion, we have moved to Materials and Methods section year.
141-142: Two PgGRAS genes were positively cor- 141 related with saponin content and negatively correlated with saponin content… Please, check.
Response: Thank you for your suggestion, we have changed to Two PgGRAS genes (PgGRAS62-03 and PgGRAS69-03) were positively correlated with saponin content, and also negatively correlated with saponin content.
191: Adjust the table to fit its content for better demonstration, i.e., column width, ..etc.
Response: Thank you for your suggestion, we have changed table to Supplementary Materials.
206: Add SD or SEM bars to the charts.
Response: Thank you for your suggestion, Figure 5. is the expression of PgGRAS68-01 gene that can add SD or SEM.
259: Check spaces.
Response: Thank you for your suggestion, we have made the changes.
281: Check the cited reference style of the journal, also it has to be consistent in the entire manuscript, “(59) (Kumari et al.), Arabidopsis Thaliana (33) (Lee et al. 281:2008), rice (57) (Tian et al. 2004), tomato (53) (Huang et al. 2015)”, these are mixed styles.
Please, check and correct in the entire manuscript.
Response: Thank you for your suggestion, we have changed the reference in this manuscript.
367: water, The aqueous solution… water. The aqueous solution…
Response: Thank you for your suggestion, we have made the changes.
481: Reference list: Check res. Titles (Sometimes all initial uppercases), also check journal names. Italicize scientific names in the entire manuscript including the ref list.
Response: Thank you for your suggestion, we have changed the reference in this manuscript.
Round 2
Reviewer 1 Report
In the original review, I mentioned that the results are frequently overstated. I hoped this would be sufficient to encourage the authors to reexamine the entire paper, however, very little has changed except for reformatting the tables. While Table S1 is now legible, it has no legend, but at least one can follow some of the argument, which is that PgGRAS68-01 is moderately correlated with ginsenoside content. This shows differing spatial expression, but not, I think, temporal. The expression of PgGRAS68-01 varies in various tissues, high in fruit and some, but not all, root samples. Over expression of PgGRAS68-01, based on fig 10, may or not significantly affect ginsenoside content (insufficient information in fig10 to tell).
The observed effect on ginsenoside level is not clearly significant. While DELLA proteins can inhibit GA in Arabidopsis, it’s not clear that is orthologous to the Arabidopsis protein in question (which is not named). In any cases, GA was not assayed so the mechanism shown in fig 11 is more than speculative; it is a complete fantasy and should be removed.
In general there is a lack of detail throughout the paper with regard to methodology, specifically, ho the significance of the correlation is determine, how the multiple alignment is done, how the GRAS gene tree is constructed, and the significance of any of the measurements in fig 10.
Abstract: the observed gene duplications do not appear to tandem in most cases, but due to the small scale of fig 1 it’s not clear. The inhibition of ginsenoside synthesis in the PgGRAS68-01 overexpressing line is ambiguous due to lack of detail in
Pg1, line 44 20s-protopanaxadiol (PPD) should be 20(S)-Protopanaxadiols, I think
Pg2, line 72-73 “while the non-conserved region is composed of intrinsically disordered domains involved in molecular recognition.” what is the evidence for this? Reference?
Table S1 is now legible but it has no legend. What do the colors indicate? From the text, the values seem to be correlation coefficients. Are they Pearson’s correlation? Matthews? Something else? Values as low as -0.305 are marked as significant (P<0.05) and one value of -0.088 marked as significant at P<0.01. This seems very unlikely; what test was used?
The authors state “Two PgGRAS genes (PgGRAS62-03 and PgGRAS69-03) were positively correlated with saponin content, and also negatively correlated with saponin content.” Pg5, line 165-167
While these genes are weakly and not significantly negatively correlated with total saponin, they are mixed with respect to gensenosides. This statement would more accurately say that they are not significantly correlated with total saponin, and are moderately correlated with some, but not all, ginsenosides. This seems like a remarkably weak reason to focus on these genes.
PgGRAS62-03 and PgGras69-03 are more clearly correlated with the enzymes shown in table 2, but not more than others such as PgGRAS62-06 or PgGRAS68-01 (and many others). Why should we focus on the two selected genes?
“the expression of 9 PgGRAS genes was significantly negatively correlated with the expression of key enzyme genes”. Pg5, line 179-180. This is overstated; the genes in question are not negatively correlated with all the key enzyme genes, and often only with one or two. More correctly, nine PgGRAS genes (list) were negatively correlated with at least one key enzyme.
Pg14,line 187-189
“results calculated by SPSS software found a total of 10 genes 187 (PgGRAS44-04, PgGRAS45-01, PgGRAS51-01, PgGRAS51-02, PgGRAS56-02, PgGRAS62- 188 06, PgGRAS63-01, PgGRAS63-02, PgGRAS65-03, and PgGRAS68-01) were significantly 189 correlated with both key enzyme genes and ginsenoside content of ginseng”. It’s really irrelevant that the test was done by SPSS, the important question is what test was performed (an how it was performed).
Pg 14, line 192
The manuscript spends considerable time discussing PgG62-03, PgG69-03 and others before deciding to focus on PgGras68-01, in spite of the fact it is not significantly correlated with either ginsenosides or total saponins. Neither is it negatively correlated with the “key enzymes” which was a big point in the earlier part of the presentation. This is very confusing. Apparently the whole discussion above was irrelevant. Fig 3, which shows a network of unknown origin, and is a poor justification for choosing to focus on PgGras68-01 since we know neither the source of the information nor whether any of the other genes are more connected to networks.
Fig4. It is entirely unclear where the secondary structur (4a) and 3D structure 4b come from. One assumes they may be predicted sturctures, but this needs to be clear. I have no idea what the lower panel of 4A is; it isn’t mentioned in the text or caption. The method used to produce the tree in 4C needs to be specified. What are the number in 4C? if these are bootstrap values, the entire tree is unresolved (i.e., all branches are very low confidence). 4D, the multiple alignment, is too small to be read. This figure is interesting to those interested in the family but all portions are too small, and no information is provided on where these results come from. The highlighted sequence in 4D is referred to as the DELLA domain, but it is actually only the DELLA sequence; the DELLA domain generally refers to the longer 20-30 residues conserved region. Given the discussion in the paper, neither the secondary structure nor the 3D structure are relevant and should be omitted. 4C would be better if all PgGras genes were included. The conclusion that g PgGRAS68-01 is closest to Os and At GRAS25 is only possible if all GRAS genes from OS and At are included (and if a tree with higher bootstrap support can be produced). Orthology can only be determined from an all against all comparison, which does not appear to be the case here.
Pg 18, lines we 220-223 “retrieved the PgGRAS68-01 gene expression data from 4 different aged stages of ginseng 221 roots, 14 different tissues of 4-year-old ginseng and 42 farmer’s cultivars of 4-year-old 222 ginseng roots”. The source this information was retrieved from is critical. Is it publically available, if not it should be included in the supplement. No URL was given for the Jilin Ginseng Unigenes database mentioned in materials and methods.
Fig 5. What are the units of expression? How is the level measured. It would be helpful if all panels had the same vertical scale
Fig 6. Is there anything in this figure different from Fig 5? The units seem to be different, why? Is there any significance to the shapes of the roots in 5C. I think this figure can be deleted.
Fig 8. Tlines 18, 19, and 30 are discussed in the text, but 18 appears to be negative and tline30 is not shown. The presence of the desired gene is important to the conclusions and should be shown. These three lines were assayed for ginsenoside content, all were found to have lower saponin and ginsenoside content. This is strange since line 18 appears to lack the gene of interest and the status of line 39 is unknown.
Fig10 has numerous problems. Is 10A total saponin content as in Table S1. What is the column labelled “contrast”? Only 10F shows a dramatic difference. Where is the statistical analysis?
